# Santin (5,7-Dihydroxy-3,6,4′-Trimetoxy-Flavone) Enhances TRAIL-Mediated Apoptosis in Colon Cancer Cells

**DOI:** 10.3390/life13020592

**Published:** 2023-02-20

**Authors:** Małgorzata Kłósek, Dagmara Jaworska, Grażyna Pietsz, Ewelina Szliszka

**Affiliations:** Department of Microbiology and Immunology, Faculty of Medical Sciences in Zabrze, Medical University of Silesia, 40-055 Katowice, Poland

**Keywords:** apoptosis, chemoprevention, colon cancer, santin activity, TRAIL

## Abstract

TRAIL (Tumor necrosis factor–Related Apoptosis-Inducing Ligand) has the ability to selectively kill cancer cells without being toxic to normal cells. This endogenous ligand plays an important role in surveillance and anti-tumor immunity. However, numerous tumor cells are resistant to TRAIL-induced apoptosis. In this study, the apoptotic effect of santin in combination with TRAIL on colon cancer cells was examined. Flow cytometry was used to detect the apoptosis and expression of death receptors (TRAIL-R1/DR4 and TRAIL-R2/DR5). Mitochondrial membrane potential (ΔΨm) was evaluated by DePsipher staining with the use of fluorescence microscopy. We have shown for the first time that flavonoid santin synergizes with TRAIL to induce apoptosis in colon cancer cells. Santin induced TRAIL-mediated apoptosis through increased expression of death receptors TRAIL-R1 and TRAIL-R2 and augmented disruption of the mitochondrial membrane in SW480 and SW620 cancer cells. The obtained data may indicate the potential role of santin in colon cancer chemoprevention through the enhancement of TRAIL-mediated apoptosis.

## 1. Introduction

Flavonoids belong to polyphenols synthesized in plants as bioactive secondary metabolites that affect their flavor, color, and pharmacological activities. Moreover, flavonoids play a significant role in the physiology of plants. By imparting color to flowers, they help attract insects. In addition, flavonoids protect plant cells from the harmful effects of ultraviolet radiation as well as perform function of plant hormones and growth regulators [1]. In humans, these compounds are associated with many health benefits which result from their broad bioactive properties, such as anti-inflammatory, anticancer, antioxidant, cytoprotective, immunomodulatory, and antimicrobial properties [2,3,4].

The structure of flavonoids is composed of two benzene rings (A and B) linked by a heterocyclic pyran ring (C-ring). According to their molecular structure, they are subdivided into different groups: anthocyanidins, flavanols, flavanones, flavonols, flavones, and isoflavones. 

Flavonoids play a major role in the human diet. The main flavonoid sources are fruits, vegetables, grains, herbs, and beverages (e.g., wine, tea, and beer) [5,6,7,8]. There is still a growing interest in the field of the research on flavonoid compounds. It was shown repeatedly, that high intake of flavonoids may provide protection against oxidation, inflammation, and consequently against cancer and chronic diseases. These compounds are able to change the activity of cell enzymes and affect the cellular systems, which can be very beneficial to the organism [2,3].

The main biological activity of these substances, which has been studied for years, is their antioxidant activity. These properties of flavonoids are structure dependent, due to the position of the B-ring on the pyran C-ring but mostly to the presence and location of hydroxyl groups in their molecule. The functional hydroxyl groups demonstrate their antioxidative properties by scavenging free radicals, chelating metal ions, but also by activation of antioxidant enzymes and inhibition of oxidases [9].

The anti-inflammatory properties have an important impact on the development of cancer. Several mechanisms have been proposed to explain anticancer properties of flavonoids. These compounds can affect the initiation and promotion stages of cancer development by, for example, induction of apoptosis, inhibiting of cell cycle, anti-proliferative effect and growth inhibition but also by affecting angiogenesis [5,10,11]. Moreover, it was demonstrated that elevated levels of ROS can be oncogenic, acting destructive on nucleic acids, proteins and lipids. Consequently it can promote genetic instability and lead to cancerogenesis. Therefore, flavonoids have been shown to inhibit tumor cell generation and proliferation by inhibition of ROS formation and reducing oxidative stress [11].

Apoptosis is a form of programmed cell death and plays a crucial role in cancer prevention. During apoptosis significant changes occur, such as nuclear chromatin condensation, DNA fragmentation and apoptotic body formation. Chemoprevention plays a significant role in inhibiting the process of carcinogenesis. Many in vitro studies have shown that flavonoids exhibit anti-cancer effects through the induction of apoptosis. This effect results from impact of flavonoids on pro- and antiapoptotic proteins, as well as on proteins that regulate cell cycle pathways. Flavonoids inhibit cell proliferation of tumor cells in the G1/S and G2/M phases of the cell cycle [12]. Either natural or synthetic flavonoids can induce apoptosis in many cancer cells. Apoptosis of cancer cells under the influence of flavones depends on the type of cells and the concentration of flavonoids used. At low concentrations, leukemias are more susceptible than solid tumors such as skin, prostate or lung cancer [13].

Compared to other flavonoids, flavones have a double bond between the second and third carbon in the flavonoid backbone. They have no substituents at the third carbon and are also oxidized at the fourth carbon [14]. Flavones are usually found in plants as *O*-and *C*-glycosides. It is estimated that an adult human takes in about 1 g of flavonoid compounds per day, mainly in the form of glycosides. In the gastrointestinal tract the deglycosylation process occurs and aglycones are absorbed in the small intestine via β-glucosidase present in the intestinal epithelium. The dietary flavonoids appear in the plasma not only as aglycones but also as sulfate or glucuronidates. Also, a significant portion of flavonoids consumed undergoes conversion by enzymes of the digestive tract microflora to simple phenols [15]. 

Santin (5,7-dihydroxy-3,6,4’-trimetoxy-flavone) is a relatively unknown compound that belongs to the flavones, which possess antimicrobial, antioxidant, anti-inflammatory, and anticancer properties. Numerous epidemiological and experimental studies provide evidence for the potential beneficial effects of flavones in neoplastic diseases. Santin is isolated from aerial parts of *Tanacetum parthenium* [16], *Tanacetum microphyllum* [17], the leaves of *Dodonaea angustifolia* [18], the buds or leaves of *Betula pubescens* and *Betula pendula* [19], and all plant parts of *Achillea cappadocica, Achillea setacea, Achillea santolinoides,* and *Achillea arabica* [20]. The compound structure is presented in Figure 1. In the available medical literature databases, little is known about the biological activity of santin.

TRAIL/Apo2L (tumor necrosis factor-related apoptosis-inducing ligand) is a cytokine that belongs to the tumor necrosis factor (TNF) superfamily. It was discovered by two independent research groups in the middle of the 1990s [21,22]. TRAIL exists as a transmembrane protein on the surface of the immune system cells: T cells, natural killer (NK) cells, natural killer T cells (NKT cells), macrophages, and dendritic cells. It can also be cleaved into a soluble form (sTRAIL) [23]. TRAIL has the ability to induce apoptosis in different types of cancer cells with no toxicity to normal cells. The pre-clinical and clinical trials show that the recombinant human TRAIL (rhTRAIL) is a promising anticancer therapeutic agent. It is well tolerated, but the therapeutic effects were often insufficient in patients with cancer because some cancer cells are resistant to apoptosis mediated by TRAIL.

Five TRAIL receptors have been identified so far. Cytoplasmic death domains (DDs) in TRAIL-R1/DR4 and TRAIL-R2/DR5, called death receptors, activate an extrinsic pathway of apoptosis in a caspase-dependent manner in cancer cells. The remaining three receptors (TRAIL-R3/DcR1, TRAIL-R4/DcR2, osteoprotegerin/OPG) are called ‘decoy receptors’ [24,25]. TRAIL induces apoptosis through binding to its death receptors TRAIL-R1 and/or TRAIL-R2. It leads to the recruitment of caspase-8 and FAS-associated protein with death domain (FADD) to the formation of functional death-inducing signaling complex (DISC). Thereafter, caspase 8 is cleaved, and it causes cleave caspase 3 and BH3-interacting domain death agonist (BID). The BID in a truncated form (tBID) translocates to the mitochondria and binds to proapoptotic proteins Bax (Bcl-2-associated X protein) and Bak (Bcl-2 homologous antagonist killer) to undergo oligomerization. It results in the formation of pores in the mitochondrial membrane, mitochondrial outer membrane permeabilization, and release of cytochrome *c* and Smac/DIABLO (second mitochondria-derived activator of caspases/direct IAP binding protein with low pI) into the cytosol. The apoptotic protease-activating factor 1 (APAF-1), cytochrome *c*, and caspase 9, along with ATP, form apoptosome, which is necessary to activate caspase 9. The caspase-9 in its active form can activate effector caspases: -3, -6, and -7 and then induce apoptosis. In type, I cancer cells, the ’extrinsic pathway,’ in which death receptors and the activation of caspase-8 are involved, is sufficient to activate caspase-3 and lead to apoptosis. In other cells named type II, the induced apoptosis requires amplification of the signal via the ‘intrinsic pathway’ involvement of the mitochondria [26]. Several mechanisms of TRAIL resistance in tumor cells have been described [27]. The cause is a decreased expression of death receptors and pro-apoptotic proteins and also an increased expression of decoy receptors and anti-apoptotic proteins in a cancer cell which leads to the insensitivity of the cytotoxic and apoptotic effects of TRAIL.

Inducing apoptosis is one of the main strategies of cancer chemoprevention. We and others have shown that natural and/or synthetic flavonoids can sensitize tumor cells to TRAIL-mediated apoptosis [28,29,30,31].

In this study, the apoptotic and cytotoxic effects of rhTRAIL combined with santin on SW480 and SW620 colon cancer cells were examined. We investigated for the first time the mechanisms by which santin enhances TRAIL/Apo2L-mediated apoptosis in cancer cells.

## 2. Materials and Methods

### 2.1. Reagents

Santin (5,7-dihydroxy-3,6,4′-trimetoxy-flavone) was obtained from Alexis Biochemicals (San Diego, CA, USA). It was dissolved in 100 mM of DMSO to obtain a final concentration equal to 0.1% (*v*/*v*) in the culture media. The tested santin’s purity was ≥98% (HPLC). Soluble recombinant human TRAIL (rhsTRAIL) was bought from PeproTech Inc. (Rocky Hill, NJ, USA).

### 2.2. Cell Culture

The experiments were conducted on SW480 and SW620 human colon cancer cells purchased from ATCC (American Type Culture Collection, Manassas, VA, USA). These colon cancer cell lines come from the same patient. The SW480 cell line was obtained from a Dukes’ type B primary adenocarcinoma of the colon (grown through the muscular layer of the large intestine but non-metastatic). The SW620 (Dukes’ type C) cell line was isolated a year later than the line SW480 and was derived from a lymph node after cancer recurred with widespread metastasis. The cells were grown in monolayer cultures in Leibovitz’s L-15 medium supplemented with 10% heat-inactivated fetal bovine serum (FBS), 100 μg/mL streptomycin, and 100 U/mL penicillin at 37 °C in a humidified atmosphere of 100% air. The reagents for the cell culture were obtained from ATCC or PAA Laboratories (Pasching, Austria). The passages were conducted twice a week. The adhered SW480 and SW620 cells were treated with a 0.25% trypsin solution. Suspensions of 0.5 × 10^6^ cells in 1 ml of medium were used for experiments [32].

### 2.3. Apoptosis Determination by Flow Cytometry with Annexin V-Fitc Staining

Apoptosis was detected by flow cytometry with the FITC Annexin V Apoptosis Detection Kit (Becton Dickinson Biosciences, San Jose, CA, USA). The SW480 and SW620 cells (0.5 × 10^6^/mL) were adhered in 24-well plates for 48 h before the experiment and then exposed to TRAIL (25–100 ng/mL) and/or santin (25 μM–100 μM) for 48 h. After this incubation, the cells treated with a 0.25% trypsin solution were washed two times with PBS (phosphate-buffered saline solution) and resuspended in 1× Binding Buffer (100 µL). Afterward, 290 μL of the cell suspension was subject to incubation with (5 μL of propidium iodide and 5 μL of annexin V-FITC for 10 min in the dark at room temperature. The population of annexin V-positive cells was analyzed by flow cytometry (LSR II, Becton Dickinson Biosciences, San Jose, CA, USA) within 1 h [33].

### 2.4. Analysis of Death Receptor Expression on the Cancer Cell Surface by Flow Cytometry

The expression of TRAIL-R1 and TRAIL-R2 receptors on the cell surface was determined by flow cytometry (LSR II, Becton Dickinson Biosciences, San Jose, CA, USA). SW480 and SW620 cancer cells (0.5 × 10^6^/mL) were adhered in 24-well plates for 48 h and stimulated with santin (50–100 μM). Accutase Solution (PAN-Biotech GmbH, Aidenbach, Germany) was used to harvest the cells, and then the cells were washed twice in PBS and resuspended in PBS containing 0.5% BSA (bovine serum albumin) [18].

### 2.5. Mitochondrial Membrane Potential

The mitochondrial membrane potentials were measured by The DePsipher Kit (R and D Systems, Minneapolis, MN) in fluorescence microscopy. SW480 and SW620 cancer cells (5 × 10^5^/mL) were seeded in a plate of 24-well for 24 h prior to the experiments. TRAIL (100 ng/mL) with or without santin (50–100 μM) was added to the cancer cells. After 48 h, the cells were washed with PBS and trypsinized to separate them from the cell culture. The cancer cells were incubated with DePsipher solution (5,5′,6,6′-tetrachloro-1,1′,3,3′-tetraethyl-benzimidazolyl carbocyanine iodide) at a concentration of 5 μg/mL for 30 min at 37 °C in the dark. After that, the cells were washed with a reaction buffer containing a stabilizer. Then cell suspension was placed on a glass slide and covered with a glass cover slip. A fluorescence inverted microscope IX51 (Olympus, Tokyo, Japan) with filter sets for FITC and TRITC was used to observe stained cells. DePsipher visualizes the potential-dependent accumulation in mitochondria, indicated by a fluorescence emission shift from red (590 nm) to green (530 nm). The cells were counted, and the number of cells with disruption of the ΔΨm was expressed as a percentage of the total cells.

### 2.6. Statistical Analysis

The data are shown as the mean ± standard deviation of three experiments performed independently in duplicate or quadruplicate (*n* = 2 or *n* = 4). The results were subject to analysis by ANOVA or Student’s *t*-test. The value of *p* < 0.05 was regarded as significant. StatSoftStatistic version 12 and Microsoft Excel 2010 were used to perform statistical analysis.

## 3. Results

We investigated the apoptotic effects of TRAIL at the concentration of 25–100 ng/mL in combination with santin at the concentration of 25–100 μM on SW480 and SW620 cancer cells. Santin significantly increased TRAIL-induced apoptosis to 42.68% ± 0.74%–73.78% ± 0.62% in SW480 cancer cells and to 39.90% ± 0.70%–93.67% ± 0.62% in SW620 cancer cells in comparison to TRAIL alone (Figure 2). Santin cooperates with TRAIL to induce apoptosis in cancer cells.

We analyzed the expression of TRAIL-R1 and TRAIL-R2 proteins in SW480 and SW620 cancer cells after 24-hour treatment with santin at the concentration of 50–100 μM by flow cytometry (Figure 3 and Figure 4).

Santin significantly increased TRAIL-R1 and TRAIL-R2 protein levels on the surface of both examined cell lines in all tested concentrations.

In order to check whether the induction of apoptosis by the combination of TRAIL and santin was mediated through death receptor TRAIL-R1 and/or TRAIL-R2, we used the TRAIL-R1/Fc and TRAIL-R2/Fc chimera proteins, which have a dominant negative function against TRAIL-R1/TRAIL-R2. The proteins blocked apoptosis caused by the co-treatment of santin with TRAIL. The obtained results suggest that santin exerted the apoptotic effect of TRAIL through the intrinsic pathway.

Changes in the permeability of the mitochondrial membrane and the loss of the mitochondrial membrane potential (ΔΨm) led to apoptosis. The intrinsic mitochondrial pathway plays a crucial role in amplifying TRAIL-induced apoptosis, and the collapse of (ΔΨm) is considered to be a hallmark of this pathway. We determined whether santin sensitizes mitochondrial dysfunction induced by TRAIL. Incubation of SW480 and SW620 cancer cells with 100 ng/ml TRAIL or 50–100 μM santin alone caused a little effect on ΔΨm (12.67% ± 1.50%, 11.89% ± 1.54% and 14.22% ± 1.56% respectively for SW480 cells and 13.11% ± 1.90%, 12.22% ± 1.39% and 14.00% ± 1.66% respectively for SW620 cells). Either TRAIL or santin treatment had little effect on ΔΨm, but when the cancer cells were subjected to the combined treatment, TRAIL with santin ΔΨm was significantly increased. The incubation of colon cancer cells with TRAIL in combination with santin resulted in a synergistic enhancement of ΔΨm loss in a large percentage of cancer cells to 55.67% ± 2.96% for SW480 cells and to 62.00% ± 4.80% for SW620 cells (Figure 5).

The results show that santin, in combination with TRAIL, is involved in the mitochondrial (intrinsic) pathway in cancer cells.

## 4. Discussion

Numerous epidemiological and experimental studies confirm the possible beneficial effects of flavonoids on human health [34]. In vitro and in vivo tests have shown that flavonoids could exert immunomodulatory and strong anticancer activities [35,36,37,38]. There are some clinical trials based on flavonoid administration in neoplastic diseases. For example, apigenin is in a phase II clinical trial in colorectal cancer treatment, and genistein is in phase I/II clinical trial in the colon and rectal cancer treatment [39]. Cancer is a heterogeneous disease that is characterized by uncontrolled growth and spread of abnormal cells which invade and metastasize to other parts of the body [40]. Flavonoids possess a wide variety of anticancer effects: they control ROS-scavenging enzyme activities, partake in arresting the cell cycle, suppress the proliferation or invasiveness of cancer cells, and induce apoptosis [41,42,43,44,45,46]. Cancer cells are resistant to apoptosis, an ordered and orchestrated death of cells that is caused by a number of signal transduction pathways as well as pro-apoptotic proteins (Bcl-2 family proteins and caspases) [35]. Apoptosis is initiated by two major pathways, i.e., extrinsic that is associated with the superfamily of tumor necrosis factor (TNF) with main signaling protein death receptors TRAIL-R1/DR4 and TRAIL-R2/DR5 and caspase 8, and intrinsic pathway, involving the mitochondria, in which Bcl-2 family proteins activate caspases 9, 3 and 7 [47,48].

Chemoprevention using naturally occurring flavonoids, due to its incidence, prevalence, and disease-related morbidity and mortality, is an attractive option in colon cancer [36,49]. Epidemiological studies have demonstrated that flavonoid intake was inversely associated with colon cancer risk [50,51]. It has already been demonstrated that the chemopreventive activity of flavonoids is exerted through the modulation of apoptotic signaling pathways [52].

Endogenous TRAIL expressed on monocytes, dendritic cells, macrophages, natural killer cells, and activated T cells is an important component of the immune defense [53]. Dysregulation of apoptotic pathways plays a significant role in the initiation and development of colon cancer [54]. Numerous in vitro tests suggest that dietary flavonoids, due to their chemopreventive advantages, are associated with the enhancement of TRAIL-induced death in cancer cells [28,29,55,56,57,58,59]. Despite the fact that the majority of colon cancer cell lines are resistant to TRAIL-mediated apoptosis, TRAIL, in combination with flavonoids, leads to the synergistic induction of cell death [60,61] Further studies on the intracellular mechanism of TRAIL-mediated apoptosis may help overcome TRAIL resistance and develop approaches to colon cancer prevention based on flavonoids.

Many in vitro studies have shown that natural and synthetic flavonoids enhance apoptosis in cancer cells. Wang et al. indicated that baicalein markedly induced apoptosis in HCT-116 colorectal cancer cells. Caspase activation are critical point in programmed cell death. By exploring the apoptotic mechanism of baicalein, the activation of caspase 3 and 9 were assayed [62]. Ha et al. found the augmentation of apoptosis in HCT-15 and HT-29 colon cancer cells through increased activation of caspase 3 and 9 and alteration of mitochondrial membrane potential and dysfunction [63]. An essential step in tumorigenesis is tumor migration and invasion. Xu M. demonstrated that apigenin at 20 and 40 µM significantly inhibited migration and invasion in SW480 colon cancer cells [64]. The anti-tumor growth and anti-metastasis effects of apigenin were obtained by Chunhua Li in three colorectal adenocarcinoma cell lines: SW480, DLD-1, and LS174T [65]. The protective effects of dietary apigenin enrichment in a chronic colitis model induced by dextran sulfate sodium in mice has been shown by Márquez-Flores Y. [66]. The studies of Xiao-Yu Ai confirm that apigenin effectively inhibits inflammatory bowel disease and colitis-associated cancer through decreased levels of the inflammatory cytokines TNF-α, IL-1β, IL-6, MCP-1, and CSF-1 and of COX-2 in tumors [67]. Moreover, the tested flavonoids inhibit the STAT3/NF-κB pathway in HCT-116 colon carcinoma cells. Another flavonoid showing chemopreventive activity against colon cancer cells is quercetin. In vitro studies have shown that quercetin reduced cell viability in a dose-dependent manner in SW480 colon cancer cells [68]. Quercetin downregulated markedly Cyclin D1 and the survivin gene in a dose-dependent manner at both the transcription and protein expression levels [68]. Dysregulation of the Wnt/β-catenin pathway plays an essential role in the early stages of colorectal carcinogenesis. Park C. demonstrated that quercetin inhibited the transcriptional activity of β-catenin/Tcf in SW480 colon cancer cells [69]. In light of available literature, little is known about the anti-cancer activity of santin. Szoka et al. have shown that santin significantly reduced the proliferation, viability, and clonogenicity of gastric cancer cells line AGS, liver cancer cells line HepG2, and colon cancer cells line DLD-1 [19]. Santin activation caspases: -3, -7, -8, -9 in cancer cells lead to apoptosis. The irreversible process of cell death determines the permeability of the membrane mitochondrial and caspase activation. Our results also have shown that the co-treatment with santin plus TRAIL causes increase in mitochondrial membrane potential in SW480 and SW620 colon cancer cells. 

The TRAIL’s receptors DR4 and DR5 have varying functions in different cancer cell types. The induction of apoptosis into colon cancer cells occurs with DR5 rather than DR4. Overexpression of DRs in cancer cells is one of the reasons for cell resistance to TRAIL. Taniguchi H. et al. have shown that baicalein up-regulates DR5 expression in SW480 colon cancer cells [70]. They used a small interfering RNA (siRNA) in suppression of this up-regulation, and it efficiently reduced the apoptosis induced by TRAIL and baicalein. They suggested that the sensitization was mediated through DR5 induction. Yoshida T. et al. showed that kaempferol markedly up-regulated DR4 and DR5 receptors in SW480 colon cancer cells [60]. They confirmed, as the previous researchers, that DR5 but not DR4 siRNA efficiently blocked induced apoptosis combined with kaempferol and TRAIL, and they indicated that DR5 up-regulation by kaempferol helps to enhance TRAIL action. In turn, a flavonolignan silibinin isolated from the *Silybum marianum* significantly increased DR4 and DR5 expressions in SW480 and SW620 cells [71]. Horinaka et al. demonstrated sensitization of colon cancer DLD-1 cells does TRAIL-mediated apoptosis by apigenin through the increased expression of DR5. A co-treatment with apigenin and TRAIL-induced the activation of caspases and Bid. Additionally, DR5/Fc chimera protein efficiently blocked the activations of Bid and caspases promoted by co-treatment with TRAIL and apigenin [72]. Ding et al. described the overcome of TRAIL-resistance colon cancer HT-29 cells by wogonin, apigenin, and chrysin by up-regulation of DR5 receptor expression [73]. Wu B. et al. have shown that luteolin alone and in combination with TRAIL significantly increased mRNA expression of death receptor 5 in non-small cell lung cancer (NSCLC) cells [74]. Apigenin also upregulated the levels of DR4 and DR5 in NSCLC cells, thus sensitizing NSCLC cells to TRAIL-induced apoptosis [75]. Min K. demonstrated that fisetin-induced DR5 expression at the transcriptional level in human renal carcinoma cells [76]. Our results have shown that santin increases the expression of death receptors DR4 and DR5 as a potential mechanism by which this flavonoid augments TRAIL-mediated apoptosis in SW480 and SW620 colon cancer cells.

## 5. Conclusions

Santin enhances TRAIL-mediated apoptosis mainly by upregulating the expression of death receptors TRAIL-R1 and TRAIL-R2 in colon cancer cells. Sensitization of cancer cells to TRAIL-induced apoptosis by santin can be a potential mechanism of their anti-tumor and chemopreventive activity and affect immune surveillance.

## Figures and Tables

**Figure 1 life-13-00592-f001:**
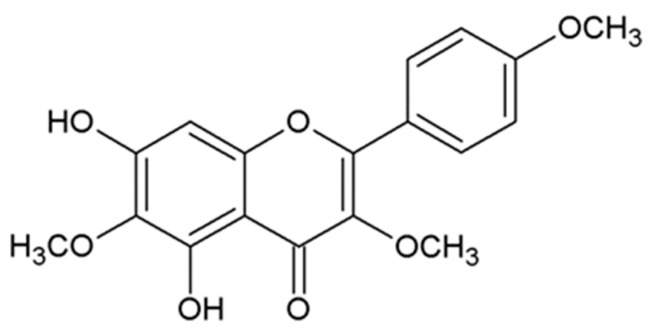
Chemical structure of santin (5,7-dihydroxy-3,6,4′-trimetoxy-flavone).

**Figure 2 life-13-00592-f002:**
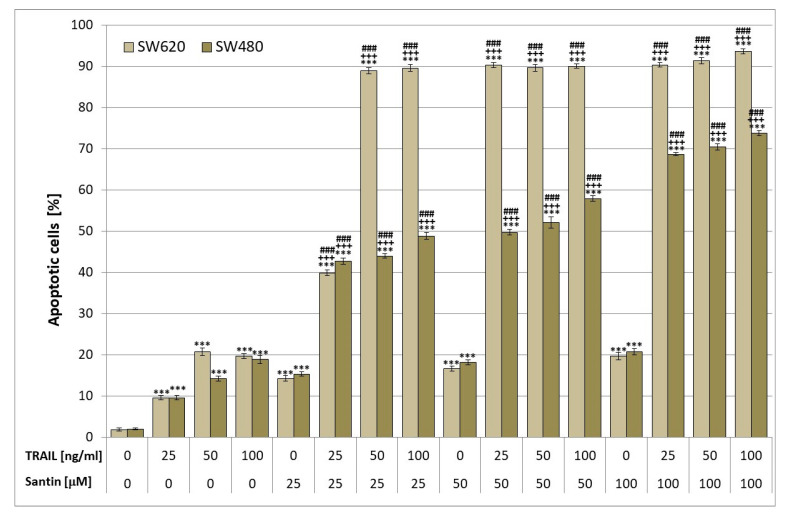
Santin enhanced TRAIL-induced apoptosis in colon cancer cells. SW480 and SW620 cells were incubated for 48 h with rhsTRAIL at the concentrations of 25–100 ng/mL and/or with 25–100 μM santin. The percentage of apoptotic cells was determined by flow cytometry using annexin V-FITC staining. The values represent mean ± SD of three independent experiments performed fourfold (*n* = 3) (*** *p* < 0.001 compared with control, ^+++^ *p* < 0.001 compared with santin, ^###^ *p* < 0.001 compared with TRAIL).

**Figure 3 life-13-00592-f003:**
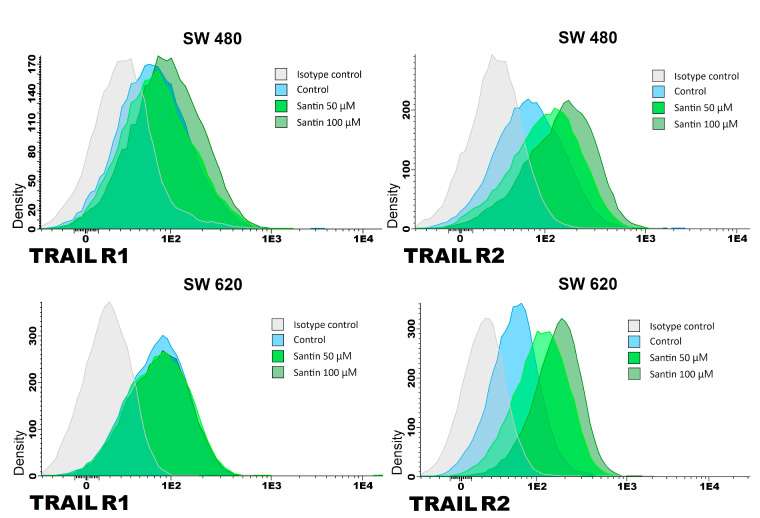
Effects of santin on the expression of TRAIL-R1 and TRAIL-R2 death receptors in SW480 and SW620 cancer cells. Cells were incubated for 24 h with santin at concentrations of 50 μM and 100 μM. The surface expression of death receptors on cancer cells was measured by flow cytometry.

**Figure 4 life-13-00592-f004:**
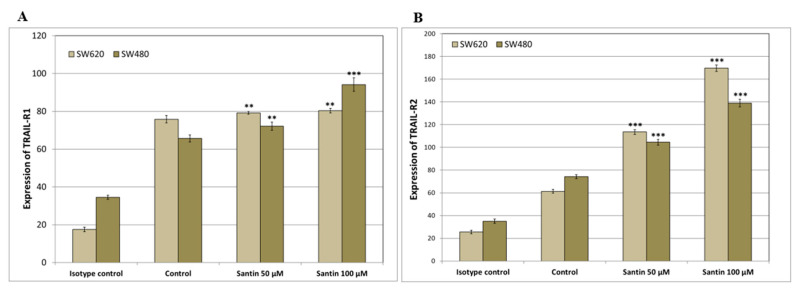
Effects of santin on the expression of TRAIL-R1 and TRAIL-R2 death receptors in SW480 and SW620 cancer cells. The values represent the mean ± SD of three independent experiments performed in duplicate (*n* = 3) (******
*p* < 0.01, *******
*p* < 0.001 compared with control).

**Figure 5 life-13-00592-f005:**
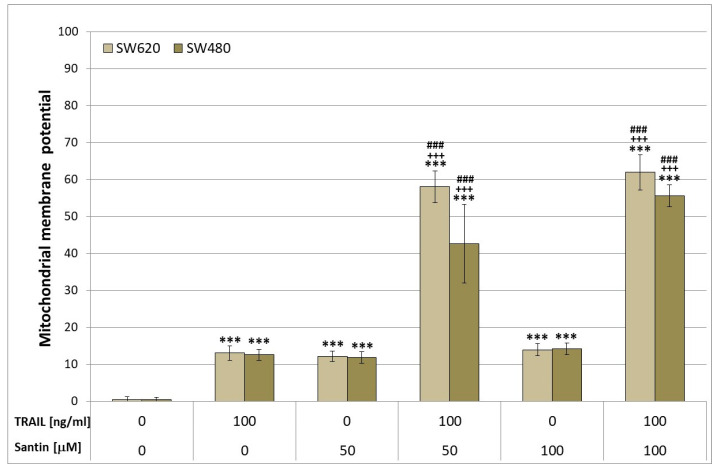
The effects of TRAIL combined with santin on the mitochondrial membrane potential (ΔΨm) in colon cancer cells. SW480 and SW620 cells were subject to incubation for 48 h with rhsTRAIL (concentration of 25–100 ng/mL) and/or with santin (25–100 μM). The fluorescent microscopic analysis of DePsipher staining was used to assess the ΔΨm loss in cancer cells (*** *p* < 0.001 compared with control, ^+++^ *p* < 0.001 compared with santin, ^###^ *p* < 0.001 compared with TRAIL).

## Data Availability

The manuscript includes all the data generated or analyzed during this study. Please contact the corresponding author to access the data presented in this study.

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
