# Peer review of "Santin (5,7-Dihydroxy-3,6,4′-Trimetoxy-Flavone) Enhances TRAIL-Mediated Apoptosis in Colon Cancer Cells"

_life, 2023, doi:10.3390/life13020592_

Round 1

Reviewer 1 Report

The images in figure 5 are not seen correctly.

The study of the use of santin as an inducer of apoptosis is novel.

However, the discussion section can be improved with recent references to compare the use of flavonoids in colon cancer chemoprevention.

Author Response

Thank you for your comments and suggestions.

Reviewer 2 Report

This manuscript is very well written, with clearly presented results and should be of interest to the basic and clinical cancer biology community.

Author Response

Thank you for your positive review and appreciation of our manuscript.

Reviewer 3 Report

The manuscript represents one of many thousands of studies on the effect of polyphenolic compounds on cancer cells. In contrast to many of them the effect of the studied compound, santin (5,7-dihydroxy-3,6,4'-trimetoxy-flavone) is quantitative and provides  good perspective for further studies. However, taking into consideration the present form of the manuscript I do not recommend its publication. However, after further experimentation and reediting it might be an interesting, small piece of experimental work worth publishing.

This is about all the paper represents.

Major comments:

1.      .The fact that it induces increased immunoreactivity of TRAIL receptors should be explored in more detail. Is this due to increased gene expression or increased stability of the protein or due to possible post translational modification?

2.      Authors use term cancer cells as an object throughout the entire text and suggest santin as an anticancer agent. However, we do not know whether this effect is specific to cancer cells, It would be ideal if they used  immortal „normal” cell line control to see whether or not there are differences in the sensitivity towards possible treatment.

3.      Similar papers have been published, some of them are not referred to. e. g. Wu B, Xiong J, Zhou Y, Wu Y, Song Y, Wang N, Chen L, Zhang J. Luteolin enhances TRAIL sensitivity in non-small cell lung cancer cells through increasing DR5 expression and Drp1-mediated mitochondrial fission. Arch Biochem Biophys. 2020 Oct 15;692:108539. doi: 10.1016/j.abb.2020.108539. Epub 2020 Aug 8. PMID: 32777260.

4.      Experiment on caspase 3 should be omitted, as it does not add anything to the theme of the paper. There should be shown the entire blot indicating procaspase-3 and cleaved caspase-3. The legend does not correspond to what is shown in the Figure. As far as this reviewer is aware, basic  caspase-3 level  is present in the cell, the level of activation changes upon apoptosis induction.

5.      The experiments on mitochondrial membrane potential should be more elaborate. The comparisons should be indicated, i.e. what is compared to what. Apart from this the number of Fig. within the text is mistaken 5 instead of 6.

6.      The data of experiment described in line 269”:”In order to check whether the induction of apoptosis by the combination of…..” should be presented in detail, as they seem important.

7.      In the Discussion and also in Introduction the stress on anti-cancer activity should be eased, there is no reason to think that this substance would ever be used as such. First of all they did not show how this compound acts on „normal” cells. The second is that phenolic substances are metabolized rapidly and their blood level is much lower compared to one used in the experiments. Please, refer to e.g. Di Lorenzo et al. Nutrients 2021, 13, 273. https://doi.org/10.3390/nu13010273.

Minor comments

The number of Figures: The results shown in Fig. 3 the authors refer to as Fig.2A-C. There are no such Figures.

Fig.2 – in the opinion of this reviewer is not necessary. It does not show any special features of these cells

Author Response

Thank you very much for detailed review of this manuscript. We thoroughly analyzed the comments posted and applied amendment.

Reviewer 4 Report

KÅ‚ósek et al. investigated if TRAIL-induced apoptosis could be restored in resistant cancer cells by also treating the cells with santin.

I have the following major concerns regarding the study:

1   1.)    The authors should have compared the effect of santin on TRAIL-induced apoptosis resistant and non-resistant cancer cells. Based on the result we could only conclude that santin is synergistic with rhsTRAIL treatment. Why were the SW480 and SW620 colon cancer cells picked for this study?

2    2.)    The authors show (Figure 4) decrease in Caspase 3 as a readout for caspase cleavage. It would have been much more precise to show the cleaved Caspase 3 product increasing upon apoptosis. That cleavage product is specifically generated upon apoptosis, while drops is Caspase 3 levels could be the result of many unrelated processes.

3   3.)    The authors claim (Figure 5) a significant increase in TRAIL-R1 levels in SW620 cells upon santin treatment. I do not see at all this increase on the FACS histograms (which perfectly overlap with each other). The authors should elaborate how they found the difference between treated and untreated cells to be even statistically significant.

Author Response

(The authors gave the same response as above.)

Round 2

Reviewer 3 Report

The authors responded to the majority of my remarks.

As caspase 3 experiment is concerned the reasoning has been improved by presentation of the entire blot. However, still, there are a lot of problems with this result.

1.      The term „expression” should not be used in this context, as the decrease of procaspase 3 is linked to the increase of „activated” caspase 3. 

2.      The effect of santin if could be seen at all should be shown ias an increase of caspase 3.

3.      The number of experiment repeats should be shown.

4.      Decimal comma should not be used in the Y axis legend.

I would also see some of the responses be included in the Discussion section.

Author Response

We are resending the responses from 20 December. The last answers from 30 December are in the 3 round

Reviewer 4 Report

I still have the following major concerns regarding the study:

11.)    The authors should have compared the effect of santin on TRAIL-induced apoptosis resistant and non-resistant cancer cells. Based on the result we could only conclude that santin is synergistic with rhsTRAIL treatment. Why were the SW480 and SW620 colon cancer cells picked for this study?

22.)    The authors show (Figure 4) decrease in Caspase 3 as a readout for caspase cleavage. It would have been much more precise to show the cleaved Caspase 3 product increasing upon apoptosis. That cleavage product is specifically generated upon apoptosis, while drops is Caspase 3 levels could be the result of many unrelated processes.

33.)    The authors claim (Figure 5) a significant increase in TRAIL-R1 levels in SW620 cells upon santin treatment. I do not see at all this increase on the FACS histograms (which perfectly overlap with each other). The authors should elaborate how they found the difference between treated and untreated cells to be even statistically significant.

Author Response

(The authors gave the same response as above.)

Round 3

Reviewer 3 Report

The Authors responded to my comments. However, as the caspase 3 experiment is concerned, they still use incorrect term „expression” concerning the existing protein. Transition of procaspase 3 into caspase 3 is the proteolytic phenomenon and has nothing to do with new gene expression. The experiment on caspase 3 seems rather inconclusive as the effect of santin on SW480 cells cannot be observed and little effect on the orher cel line can be seen.

I would recommend deletion of these results in the main body of the paper, instead, if possible I would present graphical results of the „unpublished” data which the authors mention (line 262): In order to check whether the induction of apoptosis by the combination of TRAIL and santin was mediated through death receptor TRAIL-R1 and/or TRAIL-R2, we used the TRAIL-R1/Fc and TRAIL-R2/Fc chimera proteins, which have a dominant nega-264 tive function against TRAIL-R1/TRAIL-R2 (data not shown). The proteins blocked apoptosis caused by the co-treatment of santin with TRAIL. The obtained results suggest that santin exerted the apoptotic effect of TRAIL through intrinsic pathway.”

Author Response

We thank the Reviewer for all coments and suggestions.

Reviewer 4 Report

I accept the reviewers comments and modifications on my point 2 and 3. However, regarding my first comment I have the following reply:

I like the extra information on the cell lines in the materials and methods. However, I’ll be quoting the authors from their abstract: ”We have shown for the first time that flavonoid santin overcomes TRAIL resistance in colon cancer cells”.  If the authors want to keep this claim, they need to show a comparison between cancer cells (not normal cells that do not respond to flavonoids), that are resistant to TRAIL-induced apoptosis and cancer cells that normally respond to TRAIL induced apoptosis. One would expect that santin only has an effect in the TRAIL resistant cells since the non-resistant cells will die from TRAIL-induced apoptosis even without santin. This key comparison would back up the authors claim. Otherwise, they should re-phrase their text.

Author Response

We thank for Reviewer for all comments and suggestions.
